# Effect of Acrylic Resin on the Protection Performance of Epoxy Coating for Magnesium Alloy

Xinyu Liu [1], Yingjun Zhang [1,2,*], Yong Jiang [3], Mengyang Li [1], Jianjun Bai [4] and Xiaorong Zhou [4]

[1] School of Materials Science and Engineering, Sichuan University of Science and Engineering, Zigong 643000, China; liuxysuse@163.com (X.L.); lmy97520@gmail.com (M.L.)
[2] Material Corrosion and Protection Key Laboratory of Sichuan Province, Sichuan University of Science and Engineering, Zigong 643000, China
[3] School of Chemical Engineering and Technology, Tianjin University, Tianjin 300072, China; jytl-001@163.com
[4] Sichuan Xingrong Science and Technology Co., Ltd., Meishan 620000, China; lvyande@126.com (J.B.); 18280721501@163.com (X.Z.)
* Correspondence: zhangyingjun@hrbeu.edu.cn

**Abstract:** The low toughness of epoxy resin can influence its shielding performance against a corrosive medium and strength of adhesion to metal surfaces. Extensive efforts have been made to modify epoxy resin. In this research, acrylic resin was synthesized by the solution method, and 1 wt.%, 2.5 wt.%, and 5 wt.% were added to epoxy resin (E44 brand) to prepare coatings on the surface of AZ31B magnesium alloy. The effects of acrylic resin on the mechanical and protective properties of epoxy coatings were investigated via experiments measuring impact resistance, flexibility, and adhesion as well as the electrochemical impedance technique. Compared with the pure epoxy coating, the adhesion between the coating and the substrate increases by 1.37 MPa after the addition of 2.5 wt.% acrylic resin. Meanwhile, the pencil hardness has a slight change from 5B to 6B, and the flexibility significantly improves. Therefore, the epoxy coating exhibits enhanced anticorrosive properties after the addition of 2.5 wt.% acrylic resin.

**Keywords:** acrylic resin; epoxy coating; corrosion protection; magnesium alloy





## 1. Introduction

Magnesium alloy has broad application prospects in aerospace and other fields because of the advantages of high specific strength, specific stiffness, and strong electromagnetic shielding ability. However, relatively high corrosion susceptibility limits its application range [1–4]. Some surface treatments have successfully improved the corrosion resistance of magnesium alloys, such as anodizing, microarc oxidation, and plasma electrolytic oxidation [5–7]. By contrast, organic coating is the most common choice for metal anticorrosion because of its simple process, convenient construction, and excellent protection performance.

As a typical thermoset material, epoxy resin (EP) has many advantages, such as excellent mechanical properties, effective electrical insulation capabilities, chemical corrosion resistance, and strong bonding properties [8–10]. It is often used as a primer of metallic protective coatings. However, its highly crosslinked structures and internal stress lead to low toughness in cured coatings, which can induce deformation, cracking, and stripping of substrates [11–14], resulting in microdefects or poor impact resistance of corrosion-resistance coatings. This shortcoming can influence shielding performance against a corrosive medium and strength of adhesion to metal surfaces.

Extensive efforts have been made to modify EP by adding inorganic fillers. Jing [15] introduced $MoS_2$ decorated with $ZrO_2$ nanoparticles into the epoxy coating to greatly improve the mechanical properties and corrosion resistance of the coating. Yu [16] found that grafted 3D porous graphene can improve the epoxy's tensile strength, elastic modulus,

and impact strength. Dong [17] modified cerium dioxide with fumaric acid (CeO$_2$-f) to improve its compatibility and dispersibility in EP, and coatings containing 5% CeO$_2$-f were found to exhibit optimal corrosion resistance. Other nanofillers such as SiO$_2$ [18], MOF [19], and graphite carbon nitride (g-C$_3$N$_4$) [20,21] have also been successfully incorporated into EP to improve their anti-corrosion properties. However, the poor compatibility of the inorganic fillers with organic EP is an important factor that must be addressed. Therefore, organic matter with a flexible chain structure was considered in our research.

Acrylic resin has attracted widespread attention because of its good weather resistance, corrosion resistance, durability, low price, and abundance. [22,23]. Different kinds of acrylic resin can be freely designed as needed by using different monomers and ratios. Its coatings are widely used for wall paint [24], textiles [25], paper [26], and anticorrosive coatings [27]. Some research has tried to improve the mechanical property of EP coatings by added acrylic resin. Wan [28] synthesized poly (α-methyl methacrylate-butyl acrylate-glycidyl methacrylate) (PMBG) and poly (α-methyl methacrylate-butyl acrylate) (PMB) and dispersed them in an epoxy coating through physical blending. The results indicated incorporation of 4 wt.% of acrylic resin, especially PMB, into the epoxy coating significantly enhanced the interphase adhesion of the coating (45%) through improving its inner structure. Nakamura [29] tried to reduce the internal stress generated in cured epoxy resin by shrinkage in the cooling process from cure temperature to room temperature. Three kinds of acrylic polymer were introduced by in situ ultraviolet radiation polymerization before the curing. Yu [16] found that graphene with polyacrylic acid chains adsorbed epoxy matrix molecules during stretching, creating small, localized defects in the epoxy matrix. This makes it easier for the stress effects to dissipate through these localized defects in the form of cracks, thus improving the mechanical properties of the material.

Generally, the flexibility of thermoplasticity is better than that of thermosetting. This research aimed to synthesize thermoplastic acrylic resin and add it to EP to enhance the overall performance of EP. The corrosion protection ability of EP coatings containing different amounts of acrylic resin additive in 3.5 wt.% NaCl solution on magnesium alloy was investigated. Furthermore, the mechanical properties and adhesion of the coating were tested to explore the practical applications of the coating.

## 2. Experimental

### 2.1. Experimental Materials

The raw materials used to synthesize acrylic resin monomers included butyl acrylate (BA; ChengDu Chron Chemicals Co., Ltd., Chengdu, China), methyl methacrylate (MMA; ChengDu Chron Chemicals Co., Ltd.), n-butyl acrylate (BMA; ChengDu Chron Chemicals Co., Ltd.), itaconic acid (ITA; Shanghai Macklin Biochemical Technology Co., Shanghai, China), benzoyl peroxide (BPO; Chron Chemicals Co., Chengdu, China), isooctyl 3-mercaptopropionate (IOMP, Guangzhou Swan Chemical Co., Guangzhou, China), butyl acetate (s-BAC, Chron Chemicals), and dibutyl phthalate (DBP, Kelong Chemical Reagent Co., Chengdu, China).

The E44 EP (purchased from Nantong Star Synthetic Materials Co., Ltd., Nantong, China) and LITE2015 (curing agent, purchased from Caderai Chemical Co., Ltd., Zhuhai, China) were used as coating material. The selected metal substrate was AZ31B magnesium alloy (with a chemical composition of Al 3.50%, Zn 1.40%, Mn 1.00%, Si 0.08%, Ca 0.004%, Fe 0.03%, Ni 0.01%, and the remainder Mg). The magnesium alloy was surface treated with 200- and 400-grit sandpaper, washed with deionized water and acetone, and dried for later use.

### 2.2. Synthesis of Acrylic Resin

Before polymerization, some preparatory work was completed. The initiator BPO was dissolved in DBP at a mass ratio of 1:5 and stirred at 1200 r/min for 30 min to obtain the initiator solution. The polymerization inhibitor of acrylic monomer (MMA, BA, and BMA) was removed by caustic washing. Abundant 5 wt.% sodium hydroxide solution was

added into the acrylic monomer mixture and blended by vibration. The mixture was then left to stand until stratification. The upper liquid was purified, and the monomer mixture was retained.

As shown in Figure 1, the polymerization reaction was carried out in a four-neck flask equipped with a thermometer, a condenser, a stirrer and a feed inlet. The flask was placed into a water bath, which was already adjusted to the desired reaction temperature. Subsequently, 20 wt.% purified monomer mixture and initiator, IOMP, butyl acetate, and ITA were added in a four-neck flask and mixed thoroughly. The flask was heated at 126 °C for 30 min. The remaining 80 wt.% purified monomer mixture and initiator was added dropwise within 8 h by using a micropump, and a constant temperature was maintained for 1 h. Finally, acrylic resin was obtained. After cooling, the acrylic resin was removed and stored.

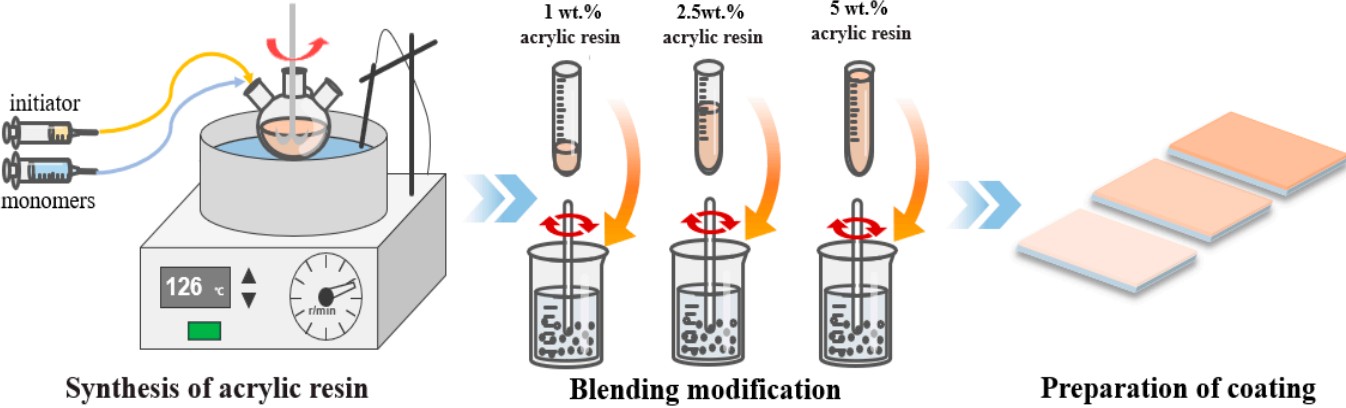

**Figure 1.** Schematic diagram of acrylic resin synthesis and coating preparation.

### 2.3. Preparation of Coatings

The polymerized acrylic resin described in Section 2.2 was dispersed in EP at 1000 r/min for 20 min to obtain a uniformly mixed resin. The amounts of acrylic resin additive were 1 wt.%, 2.5 wt.%, and 5 wt.% of the mass of the EP. The curing agent was mixed into the mixed resin according to the mass ratio of 1:1.3 (epoxy resin/curing agent), stirred evenly, and applied on the surface of the treated AZ31B magnesium alloy with a size of 5 cm × 5 cm (Figure 2). The cured thickness of the coating was 80 ± 5 μm, and a free film was simultaneously prepared on the silica gel plate. We prepared three kinds of coatings, which were denoted as 1AEP, 2.5AEP, and 5AEP. As a reference, a pure EP coating without acrylic resin was also prepared under similar conditions.

### 2.4. Characterization and Performance Testing

A Fourier-transform infrared spectrometer (FT-IR spectrometer, Platinum Elmer, Waltham, MA, USA) was used to observe the reactions between EP and acrylic resin after mixing. The purchased EP, synthetic acrylic resin, and their mixture were tested in transmission mode. Spectra were collected from 400 $cm^{-1}$ to 4000 $cm^{-1}$.

The cross-sections of the free film prepared on the silica gel plate as described in Section 2.3 were broken off and sputtered with gold for observation before SEM examination (VEGA 3SBU, Kohoutovice, Czech Republic).

In order to carry out the EIS measurements, a potential amplitude of 30 mV, peak-to-peak (AC signal) in an open circuit, was used. A frequency range between $10^5$ Hz and $10^{-2}$ Hz was adopted [30]. In order to determine the impedance parameters using simulations correlated with the experimental data obtained, a complex non-linear least squares (CNLS) simulation was utilized [30]. A CHI660 electrochemical workstation (CH Instruments, Inc., Bee Cave, TX, USA) was used to test the corrosion protection performance of the coating in 3.5 wt.% NaCl solution. The 1 $cm^2$ platinum sheet and Ag/AgCl (saturated KCl) were used as the auxiliary and reference electrodes, respectively. The coated

magnesium alloy sample prepared as described in Section 2.3 was a working electrode with a test area of 9 cm$^2$. Figure 2 is the schematic diagram of experimental apparatus.

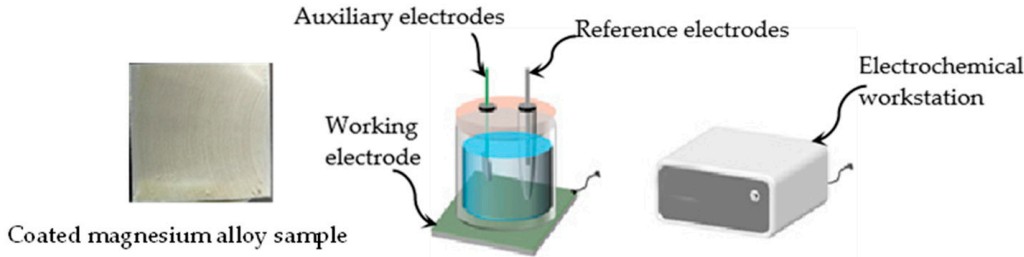

**Figure 2.** Schematic diagram of apparatus of EIS measurements.

According to the requirements of the ISO and ASTM (Table 1), the hardness, flexibility, adhesion, and contact angle of the cured coating were tested. A PPH-1 pencil hardness tester and QTX film flexibility tester produced by Shanghai Modern Environmental Engineering Technology Co., Ltd. (Shanghai, China), were utilized. The test sample was tinplate with coating according to the standard. The sample and apparatus are shown in Figure 3. The adhesion test was carried out using the BGD500 digital display pull-off adhesion tester produced by Biuged Laboratory Instruments (Guangzhou) Co., Ltd. (Guangzhou, China), and the test sample was coated magnesium alloy sample (Figure 2). The SDC-350 integral inclined contact angle measuring instrument produced by Chongqing Shengding Dyint Technology Co., Ltd. (Chongqing, China) was also employed.

**Table 1.** Test standards and instructions of the physical properties of the coating.

| Test Property | Description |  |
| --- | --- | --- |
|  | Standard | Instrument |
| Film hardness | ISO 15184:2020(E) | PPH-1 pencil hardness tester |
| Flexibility | GB/T 1731-2020 | QTX film flexibility tester |
| Adhesion | ISO 4624:2016(E) | Pull-off adhesion tester |
| Coating wettability | ASTM D5946-2004 | SDC-350 contact angle measuring device |

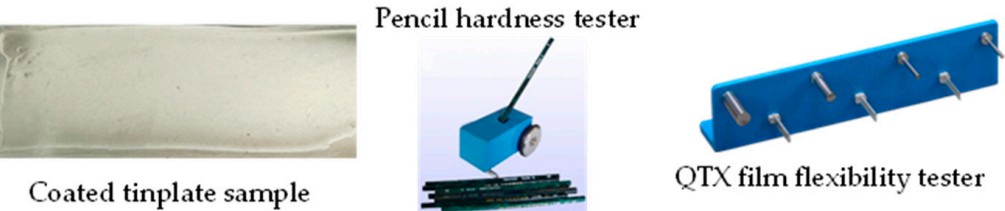

**Figure 3.** The macrograph of test sample and apparatus.

## 3. Results and Discussion

### 3.1. FT-IR Analysis of the Resin

Figure 4 presents the FT-IR spectrum of EP, synthetic acrylic resin, and their mixture. For the synthetic acrylic resin, the characteristic absorption peak that appears at 2953 cm$^{-1}$ is the stretching vibration peak of C-H [31]. The peak at 1730 cm$^{-1}$ corresponds to C=O [32,33]. The characteristic band of C-O-C of the ester group usually appears at 1300–1100 cm$^{-1}$ [34–36]. The peak at 1164 cm$^{-1}$ indicates the symmetric stretching vibration of C-O, and the peak at 1455 cm$^{-1}$ is attributed to the symmetric bending vibration peak of CH$_3$ from O-CH$_3$. Moreover, the peak at 1385 cm$^{-1}$ is due to the bending vibration of α-methyl, and the peak at 1784 cm$^{-1}$ corresponds to C-O-H. For the EP, 1608 and 1454 cm$^{-1}$ are attributed to the stretching C=C peaks of the aromatic ring [37]; 1510 cm$^{-1}$

indicates the stretching of the C-C peak of aromatic amines [38]; and 831 and 910 cm$^{-1}$ are ascribed to the stretching C-O-C peak of the oxirane group and epoxide ring vibrations, respectively [39]. For the mixed resin, however, compared with acrylic resin and EP, the characteristic absorption peaks were found in the same position, indicating that mixed resin did not undergo any chemical reactions at room temperature.

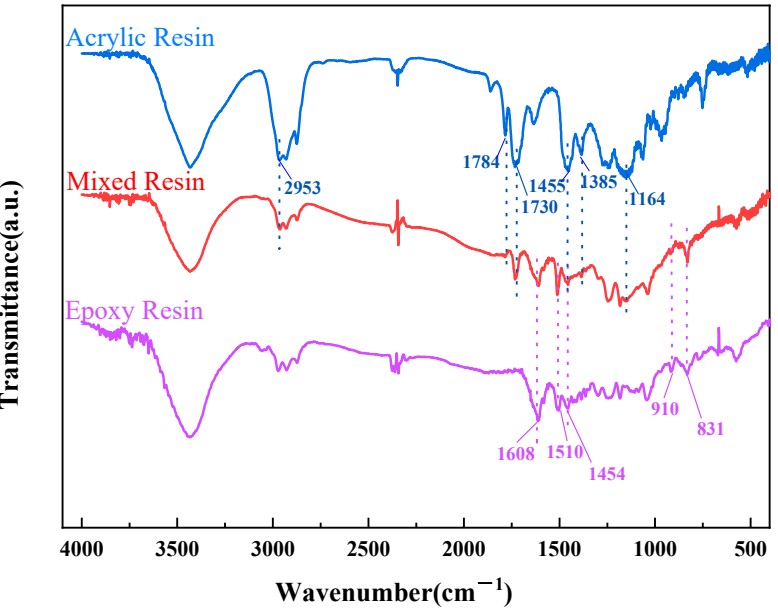

**Figure 4.** FT-IR spectrum of epoxy resin, acrylic resin, and their mixture.

### 3.2. SEM Images of Coating Section

Figure 5 shows the micro-morphology of cross-sections of the pure EP coating and coatings with different quantities of acrylic resin additive (1AEP, 2.5AEP, and 5AEP corresponding to 1 wt.%, 2.5 wt.%, and 5 wt.%, respectively). Some textures and particles were found on the fracture surface of the EP coating, resulting from breakage (Figure 5a). At the same time, some defects formed during curing were exposed. When acrylic resin was added, the textures significantly diminished, and particles obviously reduced, as shown Figure 5b–d, which indicated that the flexibility of the coatings improved. In Figure 5b, the section of the 1AEP coating is smooth, with a relatively single-crack direction, and some microplastic deformation was found on the fracture surface. Several herringbone ridge patterns were formed, and the number of textures was significantly reduced. Meanwhile, the extension paths of cracks were longer than those of the EP coating Few fine particles scattered in the fracture surface. The number of defects was significantly reduced. A similar fracture surface was found in the 2.5AEP coating (Figure 5c). Single-crack direction and long crack extension paths were formed during the sample preparation process. Several hole defects were caused by solvent volatility in the film-forming process. For the 5AEP coating, fewer textures and also fewer crack bifurcations were formed on the fracture surface (Figure 5d). The 5AEP coating had the smoothest fracture surface among test samples. None of the AEP coatings exhibited the phase separation phenomenon when acrylic resin was added. The synthetic acrylic resin was thermoplastic, with a flexible chain structure. When the AEP coating was broken, the cracks generated encountered thermoplastic phases uniformly dispersed in the resin matrix during propagation, resulting in crack deflection and prolonging the crack propagation path. A good flexibility of acrylic resin can consume a large amount of fracture energy, so the AEP coating showed a different fracture morphology.

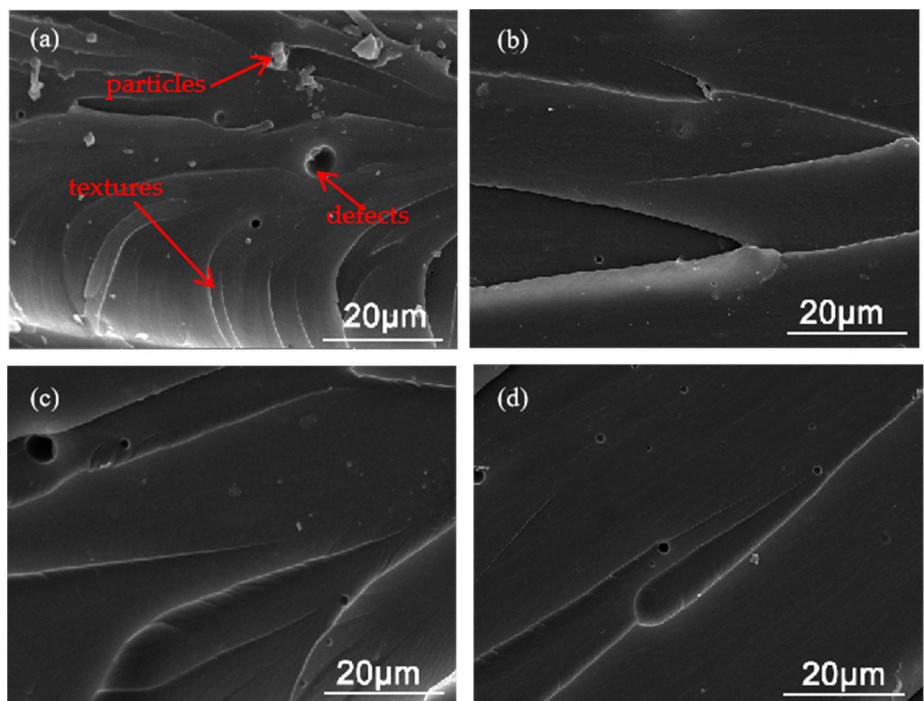

**Figure 5.** SEM of (**a**) EP coating, (**b**) 1AEP coating, (**c**) 2.5AEP coating, and (**d**) 5AEP coating.

### 3.3. Physical properties of coatings

Table 2 demonstrates the test results of some physical properties of the EP, 1AEP, 2.5AEP, and 5AEP coatings. In the flexibility test, the anti-deformability effects improved as the flexibility decreased. The value of flexibility of the EP coating was only 4 mm diameter. Compared with the EP coating, the flexibility of the 1AEP, 2.5AEP, and 5AEP coatings improved; their values were 1.0 mm, 0.5 mm, 1.5 mm radius of curvature, respectively. The smaller diameter, the better the flexibility. This result was consistent with the results of SEM. This phenomenon illustrated that the thermoplastic acrylic resin had flexible chains that could effectively resist cracking and peeling in case of force. The addition of acrylic resin led to a slight change in pencil hardness from 5B to 6B. The slight change in hardness indicated that acrylic resin had a small effect on the hardness of the EP coating. The contact angles of the 1AEP and 2.5AEP coatings were higher than those of the EP coating, but the contact angle of the 5AEP coating was lower. The contact angle is related to the surface state of the coating. The smaller the contact angle, the better its wettability, and more adverse the effect on protective performance. A suitable content of acrylic resin additives could increase the contact angle and improve the hydrophobicity of coatings.

**Table 2.** Results of physical property tests for different coatings.

| Property / Coating | Pencil Hardness | Contact Angle | Flexibility |
|---|---|---|---|
| EP | 5B | 83.8 ± 1.3° | Diameter Φ 4 mm |
| 1AEP | 6B | 91.4 ± 1.7° | Radius of curvature 1.0 ± 0.1 mm |
| 2.5AEP | 6B | 94.8 ± 1.8° | Radius of curvature 0.5 ± 0.1 mm |
| 5AEP | 6B | 80.2 ± 1.5° | Radius of curvature 1.5 ± 0.1 mm |

### 3.4. Adhesion Strength of Coatings

The effectiveness of corrosion protection provided by organic coatings is directly affected by the adhesion strength between the coating and substrate. Therefore, strong adhesion is necessary for optimal performance of the organic coating. The adhesion strength of the coating was measured by the pull-off method, which is the most commonly used test for quality control and quality assurance of organic coatings [40]. The results of adhesion strength and optical photographs after the pull-off test of the four kinds of coatings are shown in Figure 6.

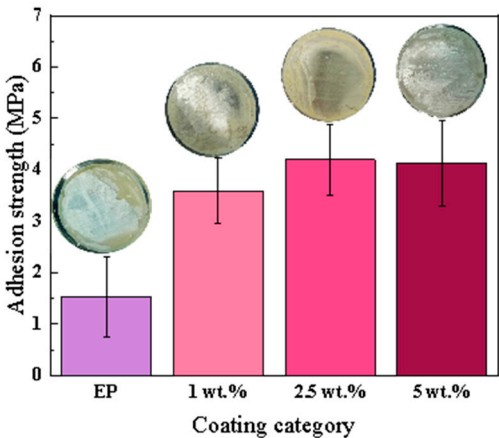

**Figure 6.** Adhesion strength of the four kinds of coatings.

The average adhesion strength for the EP coating was about 2.19 MPa. This value increased significantly when acrylic resin was added. The 2.5AEP coating had the highest adhesion strength of 3.56 MPa. The similar optical photographs after the pull-off test indicated that the EP coating and AEP coating had same failure mechanism. A large proportion of metal surfaces were exposed after the pull-off test indicated that the main failure reason was adhesion failure. Upon pulling, the coating on the surface of the metal substrate was mostly detached, and metallic luster emerged. An increase in adhesion was primarily attributed to (1) decreased curing shrinkage and (2) chemical bonding. First, the polymerized acrylic resin was thermoplastic, flexible, and resistant to shrinkage. The dispersion of acrylic resin in EP reduced curing stress and minimized negative effects on adhesion during the curing process. Second, the acrylic resin contained carboxy derived from the ITA monomer, which was also discovered in the FT-IR results in Figure 3. The carboxy group could react with $Mg^{2+}$ derived from the metallic matrix, and the formation of chemical bonds through reactions could enhance adhesion. Therefore, acrylic resin could improve the adhesion of the EP coating.

### 3.5. Analysis of Coating Protection Performance

Figure 7 displays the Nyquist and Bode diagrams of the EIS of the EP coating, 1AEP coating, 2.5AEP coating, and 5AEP coating with different immersion times. The Nyquist diagrams of EP coatings are shown in Figure 7a. They were characterized by capacitive loops at the beginning of immersion and showed an inductance semicircle after 200 h of immersion. For the Bode diagram of the EP coating (Figure 7b), a platform appeared at a low frequency range after 1 h of immersion, suggesting the presence of two time constants. Significantly, the EP coating had an inductance semicircle after 200 h of immersion. For the Nyquist diagrams of the AEP coating (Figure 7c), 2.5AEP coating (Figure 7e), and 5AEP coating (Figure 7g), only capacitive loops were observed. For the Bode diagram, the phase angle at a high frequency was close to $-90°$ for the three kinds of coatings with added acrylic resin. Thus, the coatings exhibited a high resistance value. However, the phase angle of the EP coating at a high frequency was close to $-60°$. The order of magnitude of the impedance value was slightly different. The value of the EP coating was about $10^6$ $\Omega \cdot cm^2$

at the beginning of immersion and reduced to $10^4$ $\Omega\cdot cm^2$ after 600 h of immersion. The values of 1AEP, 2.5AEP, and 5AEP coatings were $10^7$ $\Omega\cdot cm^2$, $10^8$ $\Omega\cdot cm^2$, and $10^6$ $\Omega\cdot cm^2$, respectively, which were higher than those of the EP coating at the initial immersion.

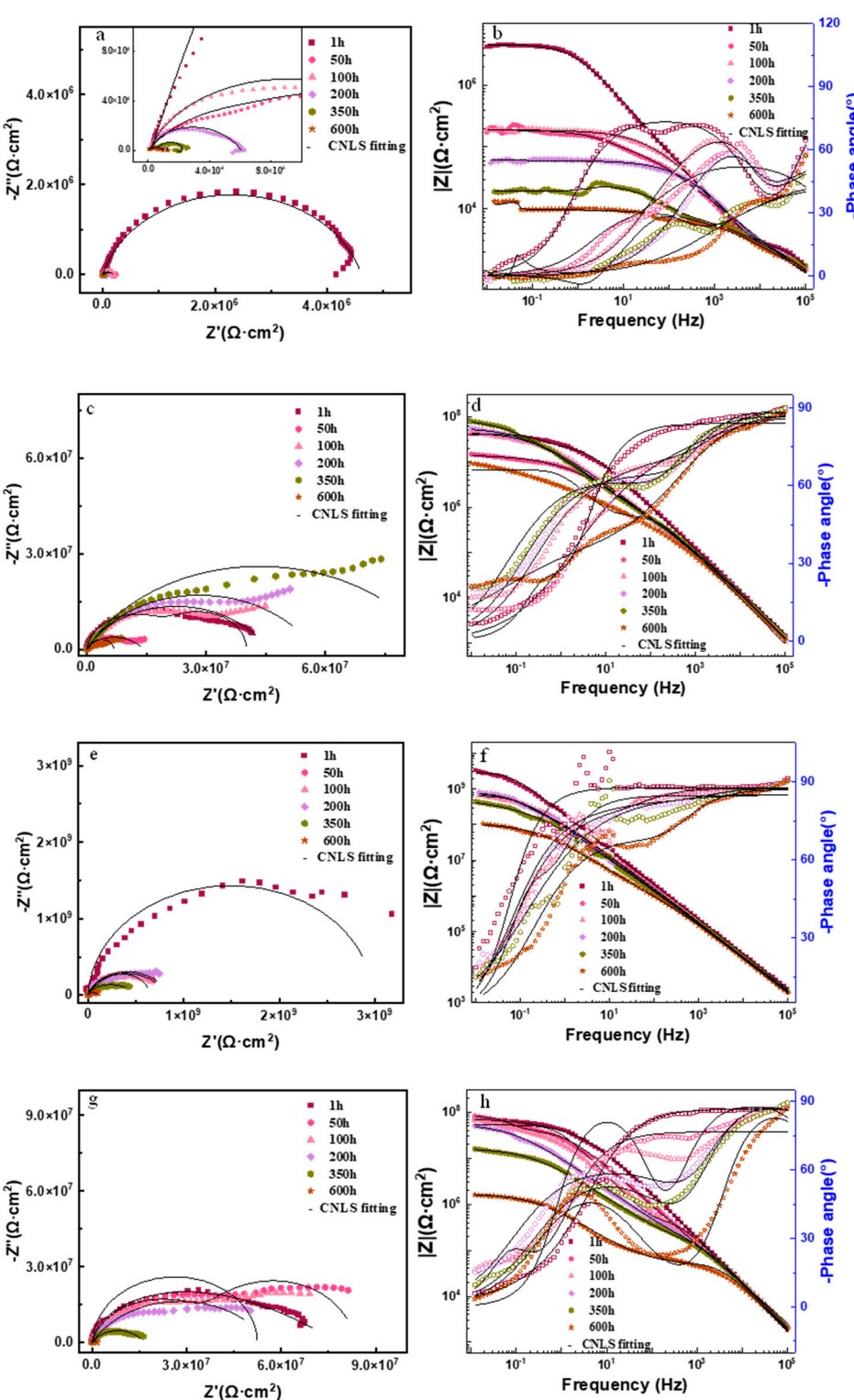

**Figure 7.** EIS spectra of EP coating (**a**,**b**), 1AEP coating (**c**,**d**), 2.5AEP coating (**e**,**f**), and 5AEP (**g**,**h**) with different immersion times.

In general, a coating has good corrosion resistance when the impedance modulus value is high at lower frequencies (0.1–0.01 Hz) [41–43]. In this study, the coating impedance

was obtained at a frequency of 0.01 Hz, as shown in Figure 8. The value of all the coatings had similar tendencies to the immersion time. The reduction in the early immersion stage could be attributed to the penetration of the solution, which reduced the shielding property of the coating. When water absorption reached saturation, or when a micropore was blocked off by water, gas, or corrosion products, the impedance value increased. The EP coating exhibited the lowest impedance value among the four kinds of studied coatings, which indicated that the acrylic resin could improve the corrosion protection performance. Among the four coatings, the 2.5AEP coating had the highest impedance value during the immersion period. The 1AEP coating and 5AEP coating had similar impedance values after 100 h of immersion, but a slight difference was observed at the beginning. These results indicated that the 2.5AEP coating provided the best protection performance for magnesium alloy compared with the other coatings.

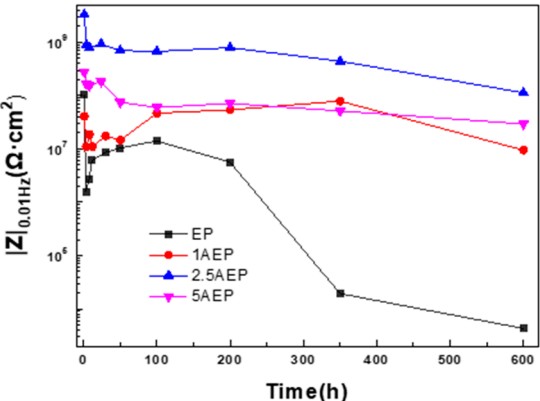

**Figure 8.** Low-frequency impedance value of the four kinds of coatings versus time.

The equivalent circuit diagram of Figure 9 was selected to fit the data, and the fitting results are shown in Figure 7 with a solid black line. For all the equivalent circuits, $R_s$ represents electrolyte resistance; $Q_c$ represents the coating capacitance; $R_{coating}$ represents the coating resistance; $Q_{dl}$ and $R_t$ represent the double-layer capacitance and charge transfer resistance of the coating-metal interface, respectively; and $L$ and $R_L$ are the inductance and inductive resistance, respectively.

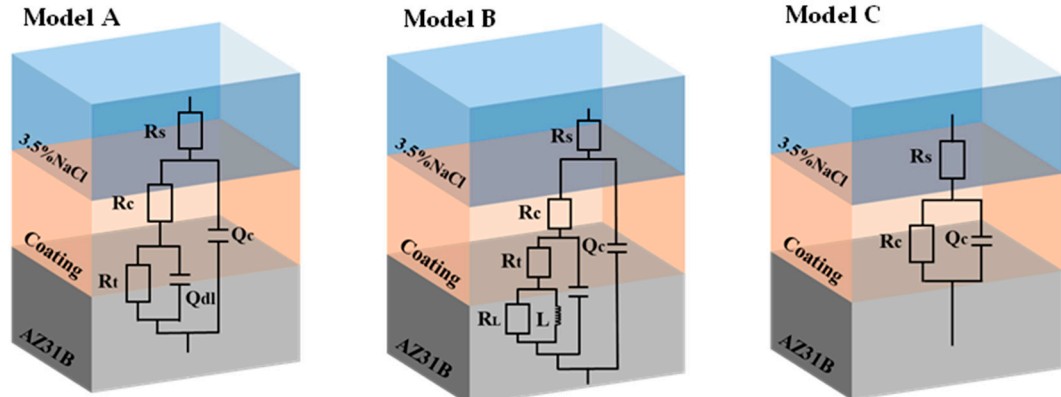

**Figure 9.** Equivalent circuit.

According to the characteristics of the EIS of the EP coating, Model A (two time constants) was used before 200 h of immersion, and Model B was used because of the inductance semicircle. The 1AEP coating and 5AEP coating at 2% had two time constants, and Model A was used. The 2.5AEP coating only had a time constant and Model C was used at the beginning of immersion; that is, there is only one capacitive semicircle in the Nyquist diagram, and there is no platform in the Bode curve in the low-frequency

region. After 8 h of immersion, Model B was selected when two capacitive semicircles were observed. The main parameter values and fitting errors are listed in Table 3.

**Table 3.** The main parameter values for different coatings.

| Coating | Immersion Time (h) | $Q_{dl}$ (F·cm$^{-2}$) | Rel. Std. Error (%) | $R_{coating}$ (Ω·cm$^2$) | Rel. Std. Error (%) | $R_t$ (Ω·cm$^2$) | Rel. Std. Error (%) |
|---|---|---|---|---|---|---|---|
| EP | 1 | $1.13 \times 10^{-9}$ | 113.2 | $2.42 \times 10^3$ | 13.05 | $4.61 \times 10^6$ | 1.39 |
| | 4 | $7.53 \times 10^{-8}$ | 9.02 | $1.66 \times 10^6$ | 7.624 | $4.04 \times 10^5$ | 69.61 |
| | 8 | $7.08 \times 10^{-8}$ | 7.21 | $2.29 \times 10^6$ | 6.411 | $6.52 \times 10^5$ | 66.77 |
| | 24 | $1.19 \times 10^{-9}$ | 105.40 | $2.00 \times 10^3$ | 12.38 | $2.25 \times 10^5$ | 0.54 |
| | 50 | $5.28 \times 10^{-7}$ | 10.08 | $6.80 \times 10^3$ | 22.82 | $1.83 \times 10^5$ | 2.95 |
| | 100 | $1.57 \times 10^{-7}$ | 6.75 | $4.89 \times 10^3$ | 11.65 | $1.83 \times 10^5$ | 1.03 |
| | 200 | $1.69 \times 10^{-7}$ | 10.79 | $5.77 \times 10^3$ | 18.03 | $5.49 \times 10^4$ | 2.14 |
| | 350 | $1.10 \times 10^{-6}$ | 99.19 | $9.31 \times 10^3$ | 35.21 | $1.02 \times 10^4$ | 28.45 |
| | 600 | $1.44 \times 10^{-6}$ | 20.10 | $9.73 \times 10^3$ | 2.90 | $1.70 \times 10^3$ | 34.93 |
| 1AEP | 1 | $2.39 \times 10^{-9}$ | 3.34 | $2.38 \times 10^7$ | 3.45 | $1.65 \times 10^7$ | 6.47 |
| | 4 | $1.39 \times 10^{-9}$ | 2.64 | $2.09 \times 10^6$ | 8.63 | $7.38 \times 10^6$ | 4.24 |
| | 8 | $3.52 \times 10^{-9}$ | 5.76 | $7.53 \times 10^6$ | 5.56 | $9.04 \times 10^6$ | 5.96 |
| | 24 | $4.86 \times 10^{-9}$ | 6.19 | $7.32 \times 10^6$ | 6.32 | $7.82 \times 10^6$ | 7.28 |
| | 50 | $2.01 \times 10^{-9}$ | 58.41 | $9.62 \times 10^4$ | 211.30 | $1.42 \times 10^7$ | 2.30 |
| | 100 | $1.64 \times 10^{-9}$ | 21.50 | $1.71 \times 10^5$ | 39.65 | $4.57 \times 10^7$ | 2.36 |
| | 200 | $1.29 \times 10^{-9}$ | 1.48 | $2.23 \times 10^5$ | 10.49 | $5.72 \times 10^7$ | 2.65 |
| | 350 | $1.95 \times 10^{-9}$ | 9.74 | $4.56 \times 10^5$ | 13.60 | $8.76 \times 10^7$ | 3.19 |
| | 600 | $1.67 \times 10^{-9}$ | 5.62 | $6.82 \times 10^5$ | 9.68 | $6.11 \times 10^6$ | 7.36 |
| 2.5AEP | 1 | $8.91 \times 10^{-10}$ | 2.915 | $2.98 \times 10^9$ | 4.75 | / | / |
| | 4 | $1.41 \times 10^{-9}$ | 3.172 | $8.45 \times 10^8$ | 3.89 | / | / |
| | 8 | $1.56 \times 10^{-9}$ | 3.284 | $7.68 \times 10^8$ | 4.20 | / | / |
| | 24 | $1.06 \times 10^{-9}$ | 5.375 | $1.85 \times 10^7$ | 64.16 | $1.13 \times 10^9$ | 4.80 |
| | 50 | $1.12 \times 10^{-9}$ | 6.416 | $1.52 \times 10^7$ | 55.91 | $8.13 \times 10^8$ | 4.82 |
| | 100 | $1.55 \times 10^{-9}$ | 3.369 | $2.50 \times 10^8$ | 9.65 | $3.95 \times 10^8$ | 7.35 |
| | 200 | $1.13 \times 10^{-9}$ | 27.43 | $1.24 \times 10^6$ | 178.40 | $8.44 \times 10^8$ | 4.98 |
| | 350 | $8.88 \times 10^{-10}$ | 4.258 | $3.29 \times 10^6$ | 22.94 | $3.30 \times 10^8$ | 6.08 |
| | 600 | $1.33 \times 10^{-9}$ | 11.28 | $8.59 \times 10^5$ | 23.30 | $1.05 \times 10^8$ | 2.98 |
| 5AEP | 1 | $1.16 \times 10^{-9}$ | 3.21 | $1.39 \times 10^7$ | 31.85 | $6.36 \times 10^7$ | 10.06 |
| | 4 | $1.24 \times 10^{-9}$ | 3.62 | $6.04 \times 10^6$ | 88.17 | $1.08 \times 10^8$ | 11.60 |
| | 8 | $1.47 \times 10^{-9}$ | 3.29 | $7.29 \times 10^6$ | 62.42 | $1.03 \times 10^8$ | 10.43 |
| | 24 | $9.24 \times 10^{-10}$ | 3.42 | $1.57 \times 10^7$ | 10.77 | $6.96 \times 10^7$ | 6.50 |
| | 50 | $4.27 \times 10^{-9}$ | 3.70 | $4.26 \times 10^7$ | 3.95 | $4.06 \times 10^7$ | 8.08 |
| | 100 | $8.74 \times 10^{-10}$ | 5.45 | $5.49 \times 10^5$ | 13.62 | $5.19 \times 10^7$ | 6.06 |
| | 200 | $8.09 \times 10^{-10}$ | 1.66 | $1.64 \times 10^5$ | 6.38 | $5.58 \times 10^7$ | 3.07 |
| | 350 | $8.28 \times 10^{-10}$ | 1.42 | $1.63 \times 10^5$ | 4.24 | $1.64 \times 10^7$ | 2.25 |
| | 600 | $9.35 \times 10^{-10}$ | 2.66 | $5.02 \times 10^4$ | 3.28 | $1.64 \times 10^6$ | 3.65 |

### 3.6. Analysis of Coating Protection Mechanism

**1.** *Protection effect of the coating*

Figure 10 shows the variation curve of the coating resistance ($R_{coating}$) of the EP coating, 1AEP coating, 2.5AEP coating, and 5AEP coating during 600 h of immersion after equivalent circuit fitting, the parameters of which are listed in Table 3. $R_{coating}$ reflects the barrier property of the coating, which is a significant element for characterizing the protective performance of the coating [44]. As a general rule, $R_{coating}$ drops rapidly at the beginning because water molecules quickly penetrate into the coating due to the relatively large dielectric constant of water. Thereafter, $R_{coating}$ fluctuates and remains at a relatively low value, resulting in a decline in the coating protective performance. Throughout the testing process, the EP coating had the lowest $R_{coating}$ value compared with the three kinds of AEP coatings. After 600 h of immersion, $R_{coating}$ of the 1AEP coating, 2.5AEP coating, and 5AEP coating was two, two, and one orders of magnitude larger than that of the EP coating, respectively. The value of $R_{coating}$ of the EP coating always remained at a low level, about $10^4$ $\Omega \cdot cm^2$, indicating a weak shielding effect against the corrosive medium. The EP coating not only formed many defects during curing (Figure 5a) but also had small contact angle (Table 2); thus, the corrosive medium was easily adsorbed on surface and diffused to the interior. The 2.5AEP coating had the highest value of $R_{coating}$ among test coatings, indicating the best shielding effect against the corrosive medium. The value changed from $10^9$ $\Omega \cdot cm^2$ at the initial immersion to $10^5$ $\Omega \cdot cm^2$ after 600 h of immersion. The 1AEP coating and 5AEP coating showed similar values at initial immersion. After 200 h of immersion, the 5AEP coating had a lower value than that of 1AEP coating. When some acrylic resin was added into the coating, its flexible chain structure could reduce the defect formation by reducing the shrinkage stress during the curing of the coating. The synthetic thermoplastic acrylic resin did not form new chemical bond linkages between coatings during the film-forming process. An increased addition of acrylic resin may cause some interfacial defects.

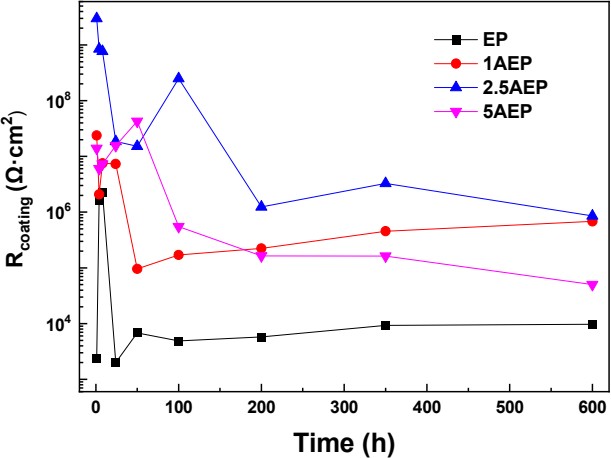

**Figure 10.** Variation curve of $R_{coating}$ with immersion time.

Figure 11 shows the variation curve of the EP coating, 1AEP coating, 2.5AEP coating, and 5AEP coating during 600 h of immersion. The charge transfer resistance ($R_t$) reflects the speed of the corrosion reaction on the electrode surface. A high $R_t$ indicates a low corrosion rate of the metal surface. At the beginning of immersion, the $R_t$ value on the electrode surface was high, making corrosion less likely to occur. The gradual decrease in $R_t$ indicated that a corrosion reaction occurred, allowing water molecules and other electrolyte solutions to pass through the coating and reach the surface of the magnesium alloy. The EP coating had the lowest $R_t$ value among all the coatings because of the lowest $R_{coating}$ value. The large $R_t$ of the 2.5AEP coating on the electrode surface suggested that the coating had relatively good protective properties, which was mainly due to the better shielding properties (Figure 10). In this case, less corrosive medium was observed on the

surface of magnesium alloy. The $R_t$ values of the 1AEP coating and 5AEP coating were in the same order of magnitude. Therefore, adding an appropriate amount of acrylic resin could improve the corrosion resistance of the EP coating.

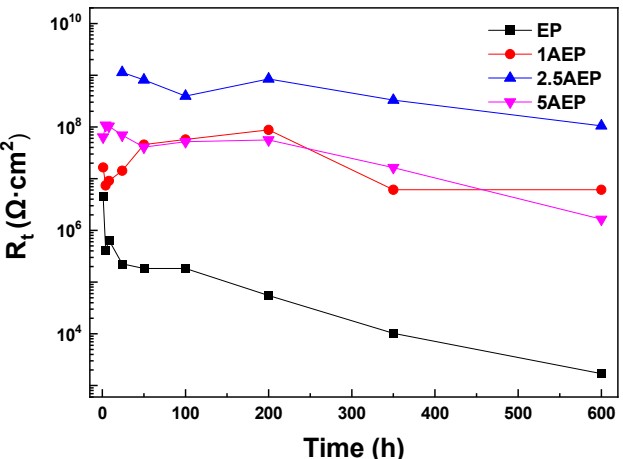

**Figure 11.** Change curve of charge transfer resistance with immersion time.

2. *Protection mechanism of coatings*

Figure 12 shows a comparison of the protection mechanisms of acrylic-resin-modified epoxy coating. For the EP coating, some defects formed during curing due to solvent volatility, and shrinkage stress occurred because of poor flexibility (Table 2). These defects provided the initial channel for solution penetration. A number of new diffusion channels formed because of water polarization and osmosis with the increase in soaking time. The contact angle was also low. Thus, the corrosive solution could rapidly permeate and diffuse through microporous structures, hence the lowest value of $R_{coating}$ for the coating (Figure 10).

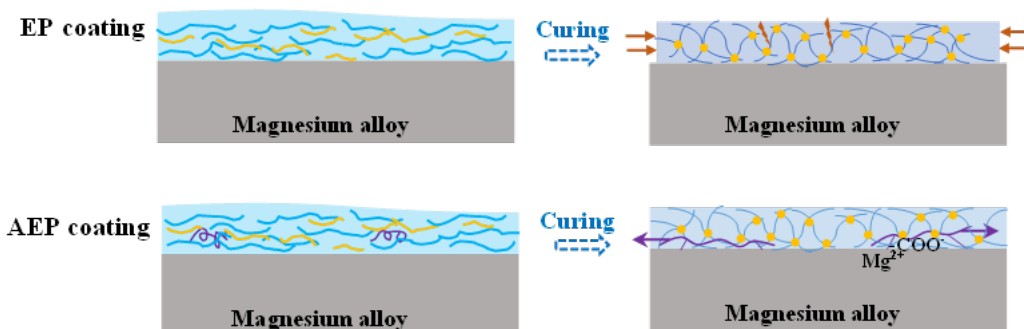

**Figure 12.** Acrylic-resin-modified epoxy coating mechanisms.

For the AEP coating, some polymerized thermoplastic acrylic resin was added and dispersed uniformly in the EP. The flexible acrylic resin chain could reduce the curing stress, which derived from the network forming process of the EP coating during the curing process. Thus, the cracks and defects in the coating were reduced, and the negative effects on adhesion force were minimized. The AEP coating had a more compact structure than the EP coating. There were few initial channels for solution penetration in the AEP coating. The high contact angle value combined with few diffusion channels gave the AEP coating a good barrier effect against the corrosive solution. Therefore, the AEP coating has a high value of $R_{coating}$ during immersion (Figure 10). The increased diffusion resistance could reduce the diffusion of the corrosive medium. The corrosion reaction on the magnesium alloy surface decreased. Meanwhile, synthetic acrylic resin containing carboxy groups could react with $Mg^{2+}$ derived from a metallic matrix. This chemical reaction could improve

the adhesion of the coating on the surface of magnesium alloy (Figure 6), thereby improving its protective performance. Therefore, the added acrylic resin could improve the corrosion protection performance of the EP coating by improving the barrier effect and increasing adhesion. But the synthetic thermoplastic acrylic resin did not form new chemical bond linkages between coatings during the film-forming process. An increased addition of acrylic resin may cause some interfacial defects. It was important to utilize an appropriate type and amount of additive.

## 4. Conclusions

In this study, the physical and corrosion protection performance of EP coatings and coatings containing three doses of acrylic resin (1 wt.%, 2.5 wt.%, and 5 wt.%) was compared. The physical properties, adhesion strength, and EIS were tested.

(1) The flexibility and adhesion strength of the EP coatings modified by acrylic resin increased, and hardness decreased.
(2) The EIS results showed that the addition of acrylic resin into the EP coating could improve the corrosion protection properties of magnesium alloys.
(3) The epoxy coating with 2.5 wt.% acrylic resin had the best corrosion resistance among the tested samples. The effects of the type and proportion of acrylic resin monomer, method of mixing, and type of additives on the coating performance will be discussed in future work.

**Author Contributions:** X.L.: Resources, Investigation, Writing—original draft. Y.Z.: Conceptualization, review and editing, Supervision. Y.J.: Methodology. M.L.: Resources, Investigation. J.B.: Resources. X.Z.: Resources. All authors have read and agreed to the published version of the manuscript.

**Funding:** This work was supported by the National Natural Science Foundation of China (No. U21A2045) and Sichuan Science and Technology Program (No. 2022NSFSC0300 and No. 2022ZHCG0076).

**Institutional Review Board Statement:** Not applicable.

**Informed Consent Statement:** Not applicable.

**Data Availability Statement:** The original contributions presented in the study are included in the article, further inquiries can be directed to the corresponding author.

**Conflicts of Interest:** Author Jianjun Bai and Xiaorong Zhou were employed by the company Sichuan Xingrong Science and Technology Co., Ltd. The remaining authors declare that the research was conducted in the absence of any commercial or financial relationships that could be construed as a potential conflict of interest.

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
