# Peer review of "Effect of Acrylic Resin on the Protection Performance of Epoxy Coating for Magnesium Alloy"

_coatings, doi:10.3390/coatings14050577_

Round 1

Reviewer 1 Report

Comments and Suggestions for Authors

The authors have applied Epoxy resin and utilized variable wt% of Acrylic resin on magnesium alloys:

(1) Authors need to correct the abstract as I didn't find any result of mechanical properties and impact resistance in the result section. 

(2) Further, any specific epoxy, acrylic resin, and magnesium alloy are included in the abstract.

(3) Why authors apply the resin coatings on the magnesium alloy, whether the corrosion behavior of the alloy is quite poor or for any specific purpose and application?

(4) The introduction section is quite weak, the authors need to build a story about why they have executed this study along with a proper literature review.

(5) The caption of Figure 1 needs to be more descriptive.

(6) Kindly include the actual images of the prepared samples, or authors may include the prepared samples for different testing as written under Section 2.4.

(7) In Figure 2, the written wavelength numbers are not clear, authors must paste a clear image here. Moreover, the description for this section is not up to the mark.

(8) Authors need to mark the obtained texture, particles, deformation, defect, etc in Figure 3 for better understanding. Further, the explanation for this section is not sufficient. This must be extended.

(9) Apart from just writing the values in Table 2, the authors should explain the obtained results of hardness, flexibility, and contact angle. Why and what these results are depicted in the direction of the objectives of the study?

 (10) How the values of pull-out 2.19 MPa and 3.56 MPa is obtained? Even, the comparative pull-out surfaces provided in Figure 4 are not clear.

(11) The legends provided in Figure 5 are not readable. Kindly paste a clear picture for the readers. The description of Figure 5 is not written well, kindly make a brief comparison of all the four types of coating behavior provided in Figure 5(a-h).

(12) How the three equivalent circuits are different as highlighted in terms of model in Figure 7. Kindly include a description of this in the revised manuscript. 

(13) What is the importance of obtaining the coating resistance, and transfer resistance as provided under Section 3.6, kindly include how these two results are important and critical as this is the only section (3.6) of the manuscript that is eye-catching.

(14) The mechanism needs further elaboration.

(15) What is the purpose of obtaining the Contact-Angle, as this is the least discussed result along with any advantage on corrosion?

(16) What is meant by impact resistance in your work? it's getting confused with the mechanical impact testing, as also written in the conclusion.

(17) Conclusion must be written in the bullet point in the sequence of presented results in the result section.

(18) Kindly include the references of the other part of the world, as it seems direction.

Comments on the Quality of English Language

Minor changes are required.

Author Response

Response to the Comments

Dear reviewer,

Thank you for giving us another opportunity to revise our manuscript entitled “Effect of acrylic resin on the protection performance of epoxy coating for magnesium alloy” (Manuscript ID: coatings-2927101). Your careful consideration and review are helpful for improving our paper and guiding our researches. We have addressed the issues in the comments point by point carefully and the corresponding revisions are also highlighted in yellow color in the revised version. Figure numbers in this letter are consistent with those in the revised manuscript. Responses the comments are as following:

Question 1: Authors need to correct the abstract as I didn't find any result of mechanical properties and impact resistance in the result section.

Response: Thanks for your suggestion and I'm sorry for this oversight. The result of mechanical properties and impact resistance had been added in the abstract. We had marked revised parts with a yellow background.

Question 2: Further, any specific epoxy, acrylic resin, and magnesium alloy are included in the abstract.

Response: Thank you for the kind suggestion. The variety of epoxy, acrylic resin, and magnesium alloy had been added in the abstract. We had marked revised parts with a yellow background.

Question 3: Why authors apply the resin coatings on the magnesium alloy, whether the corrosion behavior of the alloy is quite poor or for any specific purpose and application?

Response: There are two reasons for apply the resin coatings on the magnesium alloy. Firstly, as the lightest metal material in engineering application at present, magnesium alloy has the advantages of high specific strength, specific stiffness, and strong electromagnetic shielding ability. However, its relatively low corrosion potential and high corrosion susceptibility limits its application in many fields. Some surface treatments are necessary to improve the corrosion resistance of magnesium alloys. Secondly, organic coating (resin coating) is a commonly used method in most metal protection methods because of its simple process, convenient construction, and excellent protection performance.

Question 4: The introduction section is quite weak, the authors need to build a story about why they have executed this study along with a proper literature review.

Response:Thanks for your kind suggestion. The introduction section has been rewritten.

Question 5: The caption of Figure 1 needs to be more descriptive.

Response:Thank you for the kind suggestion. The caption of Figure 1 had modified to “Schematic diagram of acrylic resin synthesis and coating preparation”. 

Question 6: Kindly include the actual images of the prepared samples, or authors may include the prepared samples for different testing as written under Section 2.4.

Response:Thank you for the kind suggestion. The prepared samples for different testing had been added in Section 2.4. 

Question 7: In Figure 2, the written wavelength numbers are not clear, authors must paste a clear image here. Moreover, the description for this section is not up to the mark.

Response:I am sorry for this mistake. The Figure 2 had been replaced. The description for this section had been revised.

Question 8: Authors need to mark the obtained texture, particles, deformation, defect, etc in Figure 3 for better understanding. Further, the explanation for this section is not sufficient. This must be extended.

Response:Thank you for your suggestion. Some marks had been made in Figure 3 and the explanation for it had been extended and.

Question 9: Apart from just writing the values in Table 2, the authors should explain the obtained results of hardness, flexibility, and contact angle. Why and what these results are depicted in the direction of the objectives of the study?

Response:We accept this advice. As a protective layer, the coating is often subjected to various forces, so the mechanical properties of the coating are very important, and have effect on the corrosion protection performance. Therefore, these results are given. The obtained results of hardness, flexibility, and contact angle had been explained in the manuscripts.

Question 10: How the values of pull-out 2.19 MPa and 3.56 MPa is obtained? Even, the comparative pull-out surfaces provided in Figure 4 are not clear.

Response:We tested the value by Pull-off adhesion tester and refer to ISO 4624:2016(E). The 2.19 MPa and 3.56 MPa were the average value. Figure 4 has been changed.

Question 11: The legends provided in Figure 5 are not readable. Kindly paste a clear picture for the readers. The description of Figure 5 is not written well, kindly make a brief comparison of all the four types of coating behavior provided in Figure 5(a-h).

Response:Thank you for your suggestion. The pictures had been replaced. A brief comparison of all the four types of coating had been added.

Question 12: How the three equivalent circuits are different as highlighted in terms of model in Figure 7. Kindly include a description of this in the revised manuscript. 

Response:Thank you very much for your suggestion, the description of equivalent circuits has been modified.

Question 13: What is the importance of obtaining the coating resistance, and transfer resistance as provided under Section 3.6, kindly include how these two results are important and critical as this is the only section (3.6) of the manuscript that is eye-catching.

Response:Electrochemical impedance spectroscopy (EIS) is an important technology to evaluate and analyze corrosion protection performance of coating. By fitting the EIS data, the coating resistance and transfer resistance are obtained. The coating resistance generally reflects the barrier ability of a coating to an electrolyte solution. Thus, it is an important parameter for evaluating the corrosion resistance of a coating [1, 2]. The transfer resistance value is a measure of the resistance of the electron transfer across the metal surface; it is inversely proportional to the corrosion rate of metals base [3]. Therefore, the coating resistance and transfer resistance are used to analysis corrosion protection methods of coating.

[1] U. Rammelt, G. Reinhard, Application of corrosion inhibitors in water-borne coating, Prog. Org. Coat. 20 (1992) 383-392.

[2] X. W. Liu, J. P. Xiong, Y. W. Lv, Y. Zuo, Study on corrosion electrochemical behavior of several different coating systems by EIS, Prog. Org. Coatings 64 (2009) 497-503.

[3] H.H. Hassan, E. Abdelghani, M.A. Amin, Inhibition of mild steel corrosion in hydrochloric acid solution by triazole derivatives Part I. Polarization and EIS studies, Electrochim. Acta 52 (2007) 6359-6366.

Question 14: The mechanism needs further elaboration.

Response:We accept this advice. The mechanism has been elaborated further.

Question 15: What is the purpose of obtaining the Contact-Angle, as this is the least discussed result along with any advantage on corrosion?

Response:Thank you very much for your professional comment. The purpose of obtaining the Contact-Angle to observe the hydrophobic correlated with shield performance of coating. Some discussions about Contact-Angle along had been added.

Question 16: What is meant by impact resistance in your work? it's getting confused with the mechanical impact testing, as also written in the conclusion.

Response:Thank you very much for your professional comment. I am sorry to the mistake. We have replaced “impact resistance” with “flexibility” in the all section.

Question 17: Conclusion must be written in the bullet point in the sequence of presented results in the result section.

Response:Thank you very much for your suggestion. Conclusion has been written in the bullet point.

Question 18: Kindly include the references of the other part of the world, as it seems direction.

Response:Thank you very much for your suggestion and we accept it.

We sincerely hope that our response is adequate for your insightful recommendations. Thank you for the thorough review and time on our work!

Yours sincerely,

Yingjun Zhang

Reviewer 2 Report

Comments and Suggestions for Authors

The authors studied the effect of an acrylic resin on the properties of epoxy resin coatings on magnesium alloy. A concentration of 2.5% acrylic resin has been determined to be optimal for improving hardness, flexibility, adhesion and corrosion protective properties of the resin coating. An adequate co-polymerization mechanism was also proposed to explain the positive effect of acrylic resin. The work is written in a good scientific style and is well illustrated. I have the following comments and recommendations that I believe would improve the manuscript:

1. It would be better to point out the reason for selecting an AZ31B magnesium alloy as a substrate for the tested coatings. If the purpose of the coating is to protect a magnesium alloy specifically, the introduction should mention the corrosion of this magnesium alloys. 

2. How was the mixed resin coating applied on the surface of the alloy? 

3. It would be helpful if the authors could provide pictures of the resin-magnesium alloy interface.

Author Response

Dear reviewer,

Thank you for giving us another opportunity to revise our manuscript entitled “Effect of acrylic resin on the protection performance of epoxy coating for magnesium alloy” (Manuscript ID: coatings-2927101). Your careful consideration and review are helpful for improving our paper and guiding our researches. We have addressed the issues in the comments point by point carefully and the corresponding revisions are also highlighted in yellow color in the revised version. Figure numbers in this letter are consistent with those in the revised manuscript. Responses the comments are as following:

Question 1: It would be better to point out the reason for selecting an AZ31B magnesium alloy as a substrate for the tested coatings. If the purpose of the coating is to protect a magnesium alloy specifically, the introduction should mention the corrosion of this magnesium alloys.

Response:Thank you for your suggestion. The purpose of the coating is to protect many kinds of magnesium alloy, not only for AZ31B.

Question 2:How was the mixed resin coating applied on the surface of the alloy?

Response:The coating was prepared on the surface of the alloy by following two steps. Firstly, the magnesium alloy was soaked with acetone to remove the oil on the surface, then the surface of the magnesium alloy is sanded with 340 grit sandpaper, then the treated surface was wiped with anhydrous ethanol and dried with a hairdryer. Secondly, the mixed resin was uniformly applied to the surface of the magnesium alloy treated in the previous step using a customized tetrahedral preparator. We have supplied it in the manuscripts.

Question 3: It would be helpful if the authors could provide pictures of the resin-magnesium alloy interface.

Response:Thank you for your insightful advice. We will continue to improve this work in the future.

We sincerely hope that our response is adequate for your insightful recommendations. Thank you for the thorough review and time on our work!

Yours sincerely,

Yingjun Zhang

Reviewer 3 Report

Comments and Suggestions for Authors

REPORTS ON: coatings-2927101

Although the proposed manuscript has Novelty and it is reasonably organized, there are certain weaknesses, which induce to its MAJOR REVISION, as followed described:

1.                    Firstly, in the Abstract, a simple present tense should only be used. Also, English written style should be revised and improved.

2.                    In section 2, all dimensions should be accompanied with their error ranges. Besides, all experimentations should be detailed concern to duplicate or triplicate to guarantee its reproducibility. This is rather and poorly detailed/described.

3.                    Between lines 104 and 107, the follow sentences and references should be included:

“In order to carry out the EIS measurements, a potential amplitude of 10 mV, peak-to-peak (AC signal) in open-circuit with 10 points per decade is used. A frequency range between 105 Hz and 10−2 Hz is adopted [AA]. In order to determine the impedance parameters using simulations correlated with the experimental data obtained, a complex non-linear least squares (CNLS) simulation is utilized [AA].

[AA] YA Meyer, I Menezes, RS Bonatti, AD Bortolozo, WR Osório. EIS investigation of the corrosion behavior of steel bars embedded into modified concretes with eggshell contentes. Metals 202212(3), 417; https://doi.org/10.3390/met12030417

4.                    In Fig. 5, the term “fitting” should be replaced with CNLS fitting.

5.                    In Fig. 5 depict “fitting” curves, at least, it is expected that a Table with EIS parameters be shown and discussed.

6.                    If the aforementioned suggestion is neglected by authors, those CNLS fitting curves should meticulously revised and improved. This due to there are majority “simulated curves” no matching with experimental points. This indicates that “experimental” and “ simulated (CNLS)” data are erroneously selected/adopted. Based on this comment, it is clearly observed that Authors have not “KNOWN-HOW” with electrochemical analysis mainly considering equivalent circuit and CNLS analysis. Please, revise it.

7.                    Why Fig. 7 has inductance, if this phenomenon is not depicted at those experimental curves? Please, revise it, and DELETE this equivalent circuit.

8.                    Additionally, when equivalent circuit (MODEL B) is proposed, all error for each calculated term is dilute. At least “t-test” and “f-test” (regression parameters) should be provided/discussed. This “tests” demonstrates if the parameter is much larger than its standard deviation, and it presence is justified, but if it is lower, the parameter should be rejected. Please, MODEL B should be deleted and f-test” evidenced.

9.                     Figs. 8 and 9 should be revised and error ranges included.

_ _ _ _ _

Comments on the Quality of English Language

English written should be revised and improved. Please, see other aforementioned comments.

Author Response

Response to the Comments

Dear reviewer,

Thank you for giving us another opportunity to revise our manuscript entitled “Effect of acrylic resin on the protection performance of epoxy coating for magnesium alloy” (Manuscript ID: coatings-2927101). Your careful consideration and review are helpful for improving our paper and guiding our researches. We have addressed the issues in the comments point by point carefully and the corresponding revisions are also highlighted in yellow color in the revised version. Figure numbers in this letter are consistent with those in the revised manuscript. Responses the comments are as following:

Question 1: Firstly, in the Abstract, a simple present tense should only be used. Also, English written style should be revised and improved.

Response:Thank you for your comments. The tense had been altered. English written style had been revised and improved.

Question 2: In section 2, all dimensions should be accompanied with their error ranges. Besides, all experimentations should be detailed concern to duplicate or triplicate to guarantee its reproducibility. This is rather and poorly detailed/described.

Response:Thanks for your advises. All experimental result in this manuscripts were test duplicate, triplicate or multiple.

Question 3: Between lines 104 and 107, the follow sentences and references should be included : “In order to carry out the EIS measurements, a potential amplitude of 10 mV, peak-to-peak (AC signal) in open-circuit with 10 points per decade is used. A frequency range between 105Hz and 10−2Hz is adopted [AA]. In order to determine the impedance parameters using simulations correlated with the experimental data obtained, a complex non-linear least squares (CNLS) simulation is utilized [AA].

.

Response:Thanks for your advises. We have added the sentences and reference in the manuscripts.

Question 4: In Fig. 5, the term “fitting” should be replaced with CNLS fitting

Response:Thank you for your comments. The term “fitting” have been replaced with “CNLS fitting”.

Question 5: In Fig. 5 depict “fitting” curves, at least, it is expected that a Table with EIS parameters be shown and discussed.

Response:Thank you for your comments. The fitting result has been shown and discussed in Figs.8 and 9.

Question 6: If the aforementioned suggestion is neglected by authors, those CNLS fitting curves should meticulously revised and improved. This due to there are majority “simulated curves” no matching with experimental points. This indicates that “experimental” and “simulated (CNLS)” data are erroneously selected/adopted. Based on this comment, it is clearly observed that Authors have not “KNOWN-HOW” with electrochemical analysis mainly considering equivalent circuit and CNLS analysis. Please, revise it.

Response:Thank you for your professional comment and advice. In my opinion, the equivalent circuit is used following three principles: 1) Physical significance; 2) The error value of fitting result; 3) Matching with experimental points. We have more consideration on 1) and 2). We will pay more attention in future research.

Question 7: Why Fig. 7 has inductance, if this phenomenon is not depicted at those experimental curves? Please, revise it, and DELETE this equivalent circuit.

Response:Thank you for your professional comment. We have added the description of inductance in the manuscripts.

Question 8: Additionally, when equivalent circuit (MODEL B) is proposed, all error for each calculated term is dilute. At least “t-test” and “f-test” (regression parameters) should be provided/discussed. This “tests” demonstrates if the parameter is much larger than its standard deviation, and it presence is justified, but if it is lower, the parameter should be rejected. Please, MODEL B should be deleted and f-test” evidenced.

Response:Thank you for your comments. According previous researches, L and RL were the inductance and resistance which represented the Mg+ species adsorption and desorption phenomena on the surfaces of the magnesium alloy, respectively [1-3]. During the testing process, EP coating showed the inductance after 200 h immersion. Therefore, MODEL B was used.

[1] Geneviève Baril, Gonzalo Galicia, Claude Deslouis, Nadine Pébère, Bernard Tribollet, Vincent Vivier, An Impedance Investigation of the Mechanism of Pure Magnesium Corrosion in Sodium Sulfate Solutions, J. Electrochem. Soc. 154 (2) (2007) 108-113.

[2] H.W. Shi, F.C. Liu, E.H. Han, Corrosion protection of AZ91D magnesium alloy with sol-gel coating containing 2-methyl piperidine, Prog. Org. Coatings 66 (2009) 183-191.

[3] L. Kouisni, M. Azzi, F. Dalardb, S. Maximovitch, Phosphate coatings on magnesium alloy AM60. Part 2: Electrochemical behaviour in borate buffer solution, Surf. Coat. Technol. 192 (2005) 239-246.

Question 9: Figs. 8 and 9 should be revised and error ranges included.

Response:Thank you for your professional comments. The data in Figs. 8 and 9 came from the fitting result of Fig. 5. One of the coating samples. Therefore, the error ranges have been included.

We sincerely hope that our response is adequate for your insightful recommendations. Thank you for the thorough review and time on our work!

Yours sincerely,

Yingjun Zhang

Reviewer 4 Report

Comments and Suggestions for Authors

Presented paper is focused on coatings for corrosion protection of Mg alloy. Obtained results looks interesting. Nevertheless, there are several comments to paper.

With the exception of the corrosion protection part, paper lacks of discussion. Please add more discussion of obtained results.

1. Introduction. Please add more information about corrosion of Mg, and corrosion protection of this metal. Additionally, support this section with some recent papers devoted to the anticorrosion coatings for Mg and its alloy (for example, https://doi.org/10.1016/j.jma.2023.03.006, etc.).

Table 2. Please provide measurement error for contact angle.

Fig. 3. If these SEM images show cross sections, then the metal/coating bonding should be visible.

Fig. 4. Error bars are indistinguishable. Additionally, considering error values, all samples have close adhesion strength.

Fig. 5. Please provide data for bare alloy.

Please provide calculated parameters for EIS measurements.

I would recommend use “semicircle” or “loop” rather than “arc”.

Author Response

Dear reviewer,

Thank you for giving us another opportunity to revise our manuscript entitled “Effect of acrylic resin on the protection performance of epoxy coating for magnesium alloy” (Manuscript ID: coatings-2927101). Your careful consideration and review are helpful for improving our paper and guiding our researches. We have addressed the issues in the comments point by point carefully and the corresponding revisions are also highlighted in yellow color in the revised version. Figure numbers in this letter are consistent with those in the revised manuscript. Responses the comments are as following:

Question 1: With the exception of the corrosion protection part, paper lacks of discussion. Please add more discussion of obtained results.

Response:Thank you very much for your comments. Some discussion had been added in the manuscripts.

Question 2: Introduction. Please add more information about corrosion of Mg, and corrosion protection of this metal. Additionally, support this section with some recent papers devoted to the anticorrosion coatings for Mg and its alloy (for example, https://doi.org/10.1016/j.jma.2023.03.006, etc.).

Response: Thank you for your professional comment. More information about corrosion of Mg, and corrosion protection of this metal had been added in the introduction, such as “New superhydrophobic composite coatings on Mg-Mn-Ce magnesium alloy”.

Question 3: Table 2. Please provide measurement error for contact angle.

Response: Thank you for your suggestion. The measurement error for contact angle had been provided.

Question 4:Fig. 3. If these SEM images show cross sections, then the metal/coating bonding should be visible.

Response: The SEM images of cross sections of coating were obtained by free film, which was formed on the surface of silica gel plate. Therefore, the metal/coating bonding was invisible.

Question 5: Fig. 4. Error bars are indistinguishable. Additionally, considering error values, all samples have close adhesion strength.

Response:A clear picture had been used in Fig. 4. According to the ISO standards, the test result of adhesion strength comes from the average value of at least six test assemblies. The adhesion strength of EP coating was lower than AEP coating considering error values.

Question 6: Fig. 5. Please provide data for bare alloy.

Response:In previous study, we found that the value of impedance of bare magnesium alloy was very little. Indeed, there is no comparison between bare alloy and coated alloy.

Question 7: Please provide calculated parameters for EIS measurements.

Response:The calculated parameters for EIS measurements had been provided in section 2.4. “Characterization and performance testing”.

Question 8: I would recommend use “semicircle” or “loop” rather than “arc”.

Response:Thank you very much for your professional comment. The “arc” had been replaced with “semicircle”.

We sincerely hope that our response is adequate for your insightful recommendations. Thank you for the thorough review and time on our work!

Yours sincerely,

Yingjun Zhang

Round 2

Reviewer 1 Report

Comments and Suggestions for Authors

1- The introduction section is quite weak and needs to be extended according to the general guidelines for any introduction. I recommend to follow any good published manuscript for writing the introduction section. This is already indicated in Round 1 (Question 4) but the authors didn't address it.

2- For Question 3, kindly include the references that highlight  "low corrosion potential and high corrosion susceptibility" of magnesium alloys.

3- For Question 6, I didn't find any figure either in Section 2.3 or 2.4. The authors didn't address it.

4- For Question 8, no further explaination is added for Fig. 3, moreover, highlight th features for Fig 3(b-d).

5- Question 9, not addressed. Kindly inlcude the explaination and argument of obtained values in Table 2.

6- Question 10 also not addressed. Not understand what has changed in Fig.4?

7- Question 13, I didnt find any significant improvement in Section 3.6. This is  an important Section but authors didnt explained and extend it.

8- Question 14, where the mechanism is explained ans expanded?

9- Overall, a very poor revision is presented. Not acceptable revision, authors should the revision seriuously if want to proceed further.

Comments on the Quality of English Language

first authors need to revise the manuscript properly.

Author Response

Response to the Comments

Dear reviewer,

Thank you for your careful review. We have addressed the issues in the comments point by point carefully and the corresponding revisions are also highlighted in yellow color in the revised version. Responses the comments are as following:

Question 1: The introduction section is quite weak and needs to be extended according to the general guidelines for any introduction. I recommend to follow any good published manuscript for writing the introduction section. This is already indicated in Round 1 (Question 4) but the authors didn't address it.

Response: I'm very sorry for it. We had revised the introduction section again.

Question 2: For Question 3, kindly include the references that highlight "low corrosion potential and high corrosion susceptibility" of magnesium alloys.

Response: Thank you for the kind suggestion. The references 1-4 have been listed in our manuscripts.

Question 3: For Question 6, I didn't find any figure either in Section 2.3 or 2.4. The authors didn't address it.

Response: The actual images of the prepared test samples were very simple, as shown below, so we added an explanation about them in Section 2.4.

Question 4: For Question 8, no further explaination is added for Fig. 3, moreover, highlight the features for Fig 3(b-d).

Response:Thanks for your kind suggestion. We had added some explainations for Fig. 3 again.

Question 5: Question 9, not addressed. Kindly inlcude the explaination and argument of obtained values in Table 2.

Response:Thank you for your kind suggestion. We have added some explainations in this revision.

Question 6: Question 10 also not addressed. Not understand what has changed in Fig.4?

Response:We are sorry to haven't been able to solve this question very well. The macrophotograph of pull-out surfaces and Fig.4 were changed in the last revised. We have replaced it again.

Question 7: Question 13, I didn’t find any significant improvement in Section 3.6. This is an important Section but authors didn’t explain and extend it.

Response:We are sorry for this mistake. The Figure 2 had been replaced. The description for this section had been revised.

Question 8: Question 14, where the mechanism is explained and expanded?

Response:Thank you for your suggestion. The mechanism has been expanded.

Question 9: Overall, a very poor revision is presented. Not acceptable revision, authors should the revision seriuously if want to proceed further.

Response:Thank you for your suggestion. We are very sorry to haven't been able to solve these questions very well. We have revised the manuscripts again.

We sincerely hope that our response is adequate for your insightful recommendations. Thank you for the thorough review and time on our work!

Yours sincerely,

Yingjun Zhang

Reviewer 3 Report

Comments and Suggestions for Authors

Based on the rebuttals, it can be observed that the proposed manuscript was revised adequately and improved. Based on this, it deserves its final publication.

Comments on the Quality of English Language

Although it was meticulously revised, it is suggested that at proof version, a new fine revision be provided.

Author Response

Dear reviewer,

Thank you for your careful review. We have addressed the issues in the comments point by point carefully and the corresponding revisions are also highlighted in yellow color in the revised version. Respons the comment are as following:

Question 1: Although it was meticulously revised, it is suggested that at proof version, a new fine revision be provided.

Response: Thank you for the kind suggestion. We had revised the manuscripts carefully again.

We sincerely hope that our response is adequate for your insightful recommendations. Thank you for the thorough review and time on our work!

Yours sincerely,

Yingjun Zhang

Reviewer 4 Report

Comments and Suggestions for Authors

I thank authors for the careful responces for all comment. Just one remains.

Fig. 5 presents experimental points and fitting curves based on calculated parameters of equivalent circuits. Please provide these calculated parameters, i.e. Q, L, R values, etc. Ussually they are presented as table.

Author Response

Dear reviewer,

Thank you for your careful review. We have addressed the issues in the comments point by point carefully and the corresponding revisions are also highlighted in yellow color in the revised version. Response the comment are as following:

Question 1: Fig. 5 presents experimental points and fitting curves based on calculated parameters of equivalent circuits. Please provide these calculated parameters, i.e. Q, L, R values, etc. Usually they are presented as table.

Response

Question 1: Fig. 5 presents experimental points and fitting curves based on calculated parameters of equivalent circuits. Please provide these calculated parameters, i.e. Q, L, R values, etc. Usually they are presented as table.

Response: Thank you for the kind suggestion. Some calculated parameters have been added and presented as table.

Yours sincerely,

Yingjun Zhang

Round 3

Reviewer 1 Report

Comments and Suggestions for Authors

1-It is requested to include a good literature review highlighting the previous work related to your study in the introduction section.

2- Question 3 (Round 2), authors must address this "Kindly include the actual images of the prepared samples and any processing equipment, or authors may include the prepared samples for different testing as written under Section 2.2, 2.3, or 2.4.

3-Question 4 (Round 2), authors must extend the explanation of Section 3.2, also highlight the obtained features of SEM image in Fig. 3b, 3c, and 3d. This is a mandatory improvement and correction.

4- Question 6 (Round 2), the authors need to describe what these pull-out images represent above the bars and make a clear difference among the characteristics of these in Fig. 4. This is a repeating question in every round but the authors didn't address it.

 5-Question 14 (Round 2), where the mechanism is expended, kindly highlight in the revision.

6- Authors kindly take this revision seriously for proceeding further.

Comments on the Quality of English Language

minor changes.

Author Response

Dear reviewer,

Thank you for your careful review. We have addressed the issues in the comments point by point carefully and the corresponding revisions are also highlighted in yellow color in the revised version. Responses the comments are as following:

Question 1: It is requested to include a good literature review highlighting the previous work related to your study in the introduction section.

Response: Thank you for the kind suggestion. We had revised the introduction section again.

Question 2: Question 3 (Round 2), authors must address this "Kindly include the actual images of the prepared samples and any processing equipment, or authors may include the prepared samples for different testing as written under Section 2.2, 2.3, or 2.4.

Response: Thank you for the kind suggestion. The prepared samples and some equipment have added in our manuscripts.

Question 3: Question 4 (Round 2), authors must extend the explanation of Section 3.2, also highlight the obtained features of SEM image in Fig. 3b, 3c, and 3d. This is a mandatory improvement and correction.

Response: Thank you for the kind suggestion. Some explanation and description have been added again try our best.

Question 4: Question 6 (Round 2), the authors need to describe what these pull-out images represent above the bars and make a clear difference among the characteristics of these in Fig. 4. This is a repeating question in every round but the authors didn't address it.

Response:I'm sorry for not understanding in previous revision. We have added some description.

Question 5: Question 14 (Round 2), where the mechanism is expended, kindly highlight in the revision.

Response:Thank you for your kind suggestion. We have added some explaination in this revision.

Question 6: Authors kindly take this revision seriously for proceeding further.

Response:Thank you for your suggestion. We are very sorry to haven't been able to solve these questions very well. We have revised the manuscripts again.

We sincerely hope that our response is adequate for your insightful recommendations. Thank you for the thorough review and time on our work!

Yours sincerely,

Yingjun Zhang